# Multivalent contacts of the Hsp70 Ssb contribute to its architecture on ribosomes and nascent chain interaction

Marie A. Hanebuth[1,2], Roman Kityk[3], Sandra J. Fries[1,2], Alok Jain[4], Allison Kriel[5], Veronique Albanese[5], Tancred Frickey[6], Christine Peter[4], Matthias P. Mayer[3], Judith Frydman[5] & Elke Deuerling[1]

Hsp70 chaperones assist *de novo* folding of newly synthesized proteins in all cells. In yeast, the specialized Hsp70 Ssb directly binds to ribosomes. The structural basis and functional mode of recruitment of Ssb to ribosomes is not understood. Here, we present the molecular details underlying ribosome binding of Ssb in *Saccharomyces cerevisiae*. This interaction is multifaceted, involving the co-chaperone RAC and two specific regions within Ssb characterized by positive charges. The C-terminus of Ssb mediates the key contact and a second attachment point is provided by a KRR-motif in the substrate binding domain. Strikingly, ribosome binding of Ssb is not essential. Autonomous ribosome attachment becomes necessary if RAC is absent, suggesting a dual mode of Ssb recruitment to nascent chains. We propose, that the multilayered ribosomal interaction allows positioning of Ssb in an optimal orientation to the tunnel exit guaranteeing an efficient nascent polypeptide interaction.

[1] Department of Biology, Molecular Microbiology, University of Konstanz, 78457 Konstanz, Germany. [2] Konstanz Research School Chemical Biology, University of Konstanz, 78457 Konstanz, Germany. [3] Center for Molecular Biology of Heidelberg University (ZMBH), DKFZ-ZMBH Alliance, 69120 Heidelberg, Germany. [4] Department of Chemistry, University of Konstanz, 78457 Konstanz, Germany. [5] Department of Biology, Stanford University, Stanford, California 94305-5430, USA. [6] Department of Biology, Applied Bioinformatics, University of Konstanz, 78457 Konstanz, Germany. Correspondence and requests for materials should be addressed to E.D. (email: elke.deuerling@uni-konstanz.de).

The folding of newly synthesized polypeptides is a delicate and error-prone process that needs the guidance of molecular chaperones. A specialized group of ribosome-associated chaperones directly acts at the ribosomal exit tunnel where nascent chains enter the cytosol. These factors promote initial protein folding, control targeting and prevent misfolding and aggregation of newly synthesized proteins[1–6]. In eukaryotes, ribosome-associated chaperones include the nascent-polypeptide associated complex (NAC) and a specialized Hsp70/40 system (Ssb-RAC in yeast). Both systems bind transiently to the ribosome in close proximity to the polypeptide tunnel exit and interact with nascent chains[7–13].

The ribosome-bound Hsp70 Ssb is only present in fungi. In *Saccharomyces cerevisiae* it exists in two functionally interchangeable versions, Ssb1 and Ssb2, differing in four amino acids (referred to as Ssb hereafter). According to its domain structure Ssb represents a canonic Hsp70, consisting of a conserved 42 kDa N-terminal nucleotide binding domain (NBD), a 18 kDa substrate binding domain (SBD) and a 6 kDa flexible lid at the C-terminus (C-terminal domain, CTD). It transiently binds to ribosomes in a 1:1 stoichiometry, migrates with translating ribosomes and binds in a RAC-controlled manner to nascent chains[14–16]. Ssb binding to vacant ribosomes is salt-sensitive and its ribosomal interaction is not affected by ATP[11,17,18]. The ribosome-associated complex (RAC), a stable hetero-dimer of the Hsp40 Zuotin (Zuo1) and an Hsp70 Ssz1, acts as a co-chaperone for Ssb by stimulating its ATPase activity and the interaction with nascent polypeptides. RAC is conserved in higher eukaryotes and assumed to recruit cytosolic Hsp70 to nascent polypeptides[19,20]. RAC is tethered to the ribosome via Zuotin that contacts both ribosomal subunits, 60S and 40S, thereby bridging two important functional regions, the peptidyltransferase centre and the ribosomal exit, which may allow to link protein translation with early nascent chain folding[21]. The exact role of the Ssb-RAC system during protein biogenesis remains elusive but a regulatory role not only in protein folding but also in the control of translation is hypothesized[17,22–24]. The loss of Ssb-RAC enhances read through of stop codons, while the overexpression of Ssb allows their efficient recognition[25,26]. Furthermore, deletion of Ssb-RAC leads to inhibition of -1 programmed ribosomal frameshifts likely due to impaired chaperoning, which provokes backup of the nascent chain within the tunnel and aa-tRNA mispositioning[27]. In general, cells lacking individual components or the whole Ssb-RAC system show a similar pleiotropic phenotype including slow growth, sensitivity towards salt, aminoglycosides, translation inhibitory drugs, protein folding stress and low temperature. In the absence of Ssb many proteins, including ribosomal proteins and ribosome biogenesis factors, aggregate resulting in decreased amounts of ribosomal subunits and reduced translation activity[11,16,28–31].

The mechanism by which Ssb binds to the ribosome is enigmatic, while ribosome attachment of other chaperones such as bacterial Trigger Factor or eukaryotic RAC and NAC are well characterized. Moreover, it is unclear whether ribosomal interaction of Ssb is a prerequisite for its *in vivo* function, as it is the case for Trigger Factor or NAC (refs 31,32). An earlier study using domain chimeras of Ssb and another Hsp70, Ssa, revealed that neither the NBD nor the SBD of Ssb are essential for ribosome attachment[14], but there is so far no detailed molecular understanding of Ssb binding to the translation machinery.

In this study we clarify this long-standing open question by identifying the elements of Ssb that mediate ribosomal interaction and by analysing the functional consequences for a ribosome-binding mutant of Ssb *in vivo*. The interaction of this Hsp70 with the ribosome involves two Ssb-specific positively charged regions, one in the SBD and one in the CTD that mediate direct contacts to the translation machinery. The cooperation of Ssb with RAC represents a further layer of interaction and the presence of this co-factor becomes essential if a ribosome-binding mutant of Ssb is analysed. These multilayered ribosomal interactions may position Ssb in an optimal orientation at the tunnel exit to allow efficient interaction with the nascent chain.

## Results

**Ssb contains basic regions that are not conserved in Hsp70s.** The four Hsp70 subfamilies of *S. cerevisiae* (Ssa1–4, Ssb1–2, Sse1–2 and Ssz1; refs 33,34) show a high degree of conservation and especially the abundant Ssa and Ssb chaperones are more than 60% identical[35]. Nevertheless, they fulfil different tasks within the cell; Ssb is the major Hsp70 interacting with ribosomes[14]. Taking this into account, we aligned the sequences of Ssa1 and Ssb1 and screened for differences between both. Since it is known that bacterial Trigger Factor[32], as well as eukaryotic NAC (ref. 31) and RAC (ref. 24) mainly involve positive charge to contact the ribosome, we focused on differences in positively charged residues, which might be involved in the direct interaction between Ssb and the ribosome. We identified two regions in the SBD and CTD (Fig. 1a, grey boxes), which differ between Ssa and Ssb and show an accumulation of basic residues in Ssb. The first region KRR428–30 lies within a beta-sheet of the SBD, facing to the outside, not to the putative substrate-binding pocket of a modelled Ssb structure (Fig. 1b). This KRR-stretch is conserved within all Ssbs but not present in other Hsp70s. With a net charge of $+3$ this region clearly differs from the corresponding KSE421-3 residues of Ssa1 with a net charge of $\pm 0$. The second region we identified is located at the very C-terminus of Ssb (603–613) and includes 4 positively charged residues (K603, R604, K608, R613; Fig. 1a). The analogous residues within Ssa1 (595–605) include a negatively (D596) as well as one positively (K604) charged amino acid, resulting again in a net charge of $\pm 0$.

**The C-terminus of Ssb is essential for ribosome binding.** First, we analysed the impact of the C-terminus of Ssb on ribosome binding and deleted 13 residues from the CTD, resulting in the Ssb1Δ601–13 mutant (Fig. 2a). This Ssb variant, when expressed from plasmid under control of the authentic Ssb promoter, fully complemented growth of *ssb1,2Δ* cells under all conditions tested (cold, salt, translational or protein folding stress; Fig. 2b). We conclude, that the expression of Ssb1Δ601–13 supports growth of yeast cells similarly well as wt Ssb.

Next, we did a ribosome sedimentation assay to directly investigate the association of Ssb1Δ601–13 with ribosomes. Rpl17A served as ribosomal and Pgk1 as cytosolic marker. The majority of chromosomally or plasmid encoded Ssb1 was found in the ribosomal pellet (Fig. 2c). Strikingly, the mutant Ssb1Δ601–13 shifted completely into the cytosolic fraction indicating the loss of its ribosome binding capability.

To investigate whether deletion of all 13 C-terminal amino acids is necessary to abolish ribosome binding, we C-terminally truncated Ssb progressively, by deleting only four (Ssb1Δ610–13) or eight (Ssb1Δ606–13) residues (Supplementary Fig. 1a). All Ssb1 mutants complemented growth of *ssb1,2Δ* cells, as well as wt Ssb1 (Supplementary Fig. 1b). Analysing association of these deletion variants with ribosomes revealed that the deletion of only four residues already hampered ribosome binding and the loss of eight amino acids enhanced this defect even further (Supplementary Fig. 1c). Interestingly, the incubation of lysates with puromycin reduced the level of all Ssb1 versions that bind to ribosomes even more, indicating that an interaction with the nascent chain plays a role in all cases (Supplementary Fig. 1d).

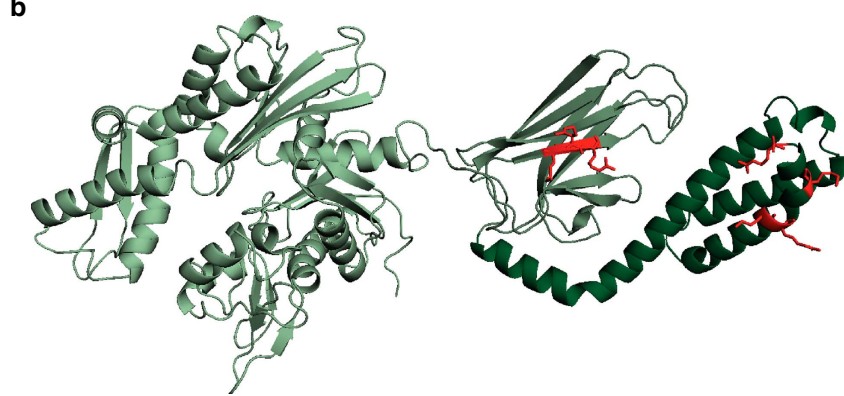

**Figure 1 | Two positively charged regions that differ between the yeast Hsp70s Ssa and Ssb might be involved in ribosome binding of Ssb.**
(**a**) Alignment of the substrate binding domain (SBD) and the C-terminal domain (CTD) of the yeast Hsp70s Ssa1 (upper lane) and Ssb1 (lower lane). Residues were marked according to their character: neutral/non-polar (green), neutral/polar (yellow), basic (blue), acidic (red); residues in black are identical in Ssa1 and Ssb1 or belong to the same group of amino acids as determined above. Grey boxes mark patches that differ between Ssa1 and Ssb1 and show a basic character in Ssb1. (**b**) Structural model of Ssb1 based on a DnaK structure (with ADP and substrate;[49]) showing the N-terminal nucleotide binding domain in light green, the SBD in middle green and the CTD in dark green. Basic residues (K, R) which are marked as grey boxes in **a** are highlighted in red.

To test ribosome binding *in vitro*, recombinantly purified wt Ssb1 and Ssb1Δ601–13 protein was incubated with puromycin stripped and high salt washed yeast ribosomes (Supplementary Fig. 2a) that were almost completely devoid of exit site associated factors like NAC or RAC (Supplementary Fig. 2b). Full length Ssb1 showed a ribosomal interaction as it shifted from the supernatant into the pellet fraction (Supplementary Fig. 2c). In contrast, Ssb1Δ601–13 did not bind to purified ribosomes, even when incubated in eightfold excess and under physiological salt conditions (Supplementary Fig. 2d). Thus, deletion of the 13 C-terminal residues of Ssb completely abolishes ribosomal interaction *in vivo* and *in vitro* supporting the assumption that this C-terminal lid element is essential for ribosome binding.

Molecular dynamics (MD) simulations of yeast Hsp70s showed in the case of Ssa a clear interaction between the SBD and the CTD, and a lid closing movement of the C-terminus with time (Supplementary Fig. 3a). This interaction was not that significantly detectable for a wt Ssb model, which did not show a strong interaction between the SBD and the CTD (Supplementary Fig. 3b), supporting the hypothesis that this region in Ssb might directly be involved in ribosome binding. However, MD simulations also suggest that deletion of the 13 C-terminal residues of Ssb even more weakens the interaction of the helical lid with the beta-sheet domain (Supplementary Fig. 3c). Thus, we cannot exclude that local misfolding of the helical CTD also contributes to the loss of ribosome binding. To test this further, we introduced point mutations in the C-terminal domain instead of deleting this region. Since we speculated that the interaction of Ssb with ribosomes is mediated via electrostatic interactions, we

substituted the four positively charged lysine and arginine residues within Ssb603–13 by alanines (Ssb1_KR603AA-K608A-R613A) creating the mutant Ssb1_KRKR-AAAA (Supplementary Fig. 4a). Alanine substitutions not only diminish the positive charge but are also known to have high helix propensity and thus should prevent potential unfolding of the lid. The expression of this Ssb1 variant with a replacement of positive to neutral charge within the C-terminus complemented the growth defects of *ssb1,2Δ* cells (Supplementary Fig. 4b), but led to a shift of Ssb1 from translating ribosomes into the soluble fraction (Supplementary Fig. 4c). Thus, we conclude that electrostatic interactions of the Ssb CTD are crucial for ribosome attachment and that the ribosome-binding defect of Ssb1Δ601–13 is mainly due to the lack of positive charges rather than local unfolding of the CTD. Since the C-terminal truncation of Ssb1 lacking only one (Ssb1Δ610–13) or two (Ssb1Δ606–13) basic amino acids already strongly influenced ribosome binding we assume that at least two out of four basic residues are crucial to mediate the ribosomal contact (Supplementary Fig. 1c).

**The KRR-motif contributes to ribosome binding.** Next, we focused on the second region KRR428–30, which we identified as a second potential ribosomal contact site of Ssb. Hypothesizing, that again electrostatic interactions play a role in this putative contact we exchanged the three lysine and arginine residues to alanines creating the Ssb mutant Ssb1_KRR–AAA (Fig. 2a). This mutant fully complemented the growth defect of *ssb1,2Δ* cells under normal and stress conditions (Fig. 2d). In a ribosome

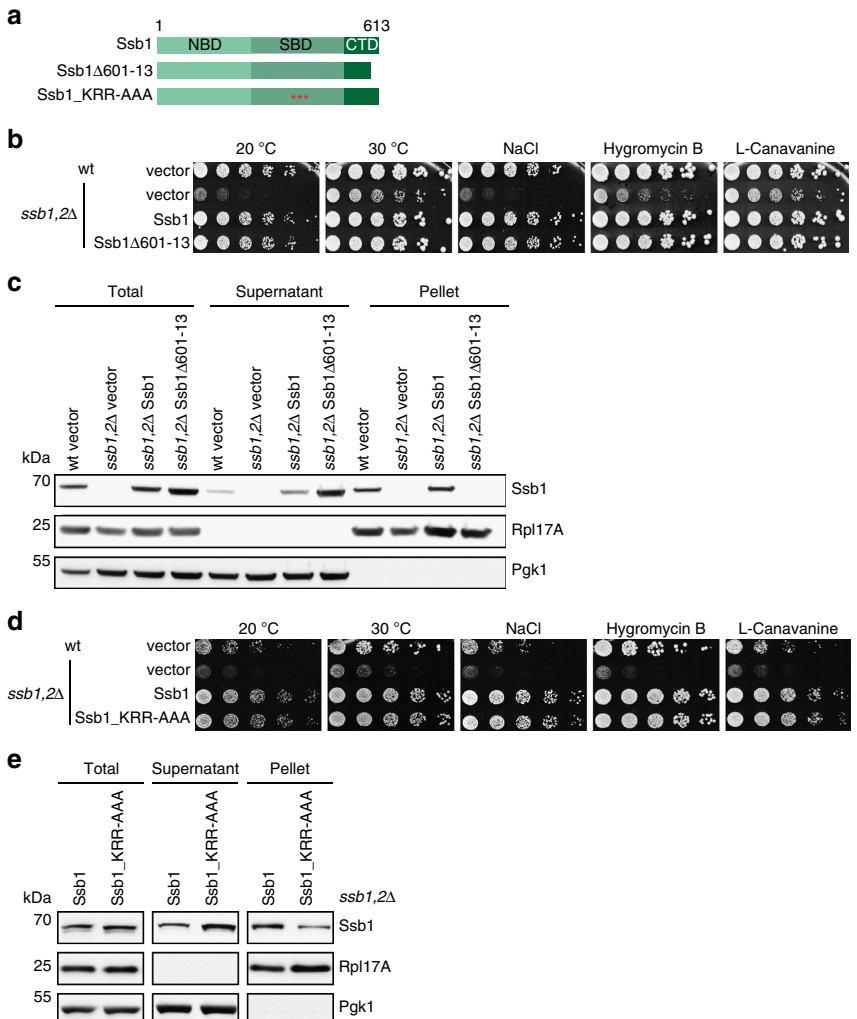

**Figure 2 | Mutations that influence ribosome binding of Ssb do not hamper growth.** (**a**) Schematic overview of Ssb domains and the constructs used. Ssb1Δ601–13 lacks 13 C-terminal residues; red asterisks mark point mutations (Ssb1_KRR428AAA). (**b**) Growth analysis of Ssb C-terminal deletion mutant. Wild type (wt) or *ssb1,2*Δ cells transformed with either empty vector or different Ssb1 versions were adjusted to $OD_{600} = 0.4$ and spotted in fivefold serial dilutions onto SC-URA plates (that may contain certain additives), which were incubated at 30 °C for two days or at 20 °C for five days. (**c**) Ribosome sedimentation assay to investigate ribosome binding of Ssb1Δ601–13. Cells were grown to exponential growth phase in SC-URA media and lysates were adjusted to same $A_{260}$ levels. 1.5 $A_{260}$ units were loaded onto a 25% (w/v) sucrose cushion followed by ultracentrifugation to pellet ribosomes. Equal amounts of adjusted totals, supernatant and ribosomal pellet fraction were analysed via SDS–PAGE followed by Western blotting. (**d**) Growth analysis of Ssb1_KRR428AAA mutant as described in **b**. (**e**) Ribosome sedimentation assay as described in **c** to analyse ribosome binding of Ssb1_KRR428AAA mutant.

sedimentation assay Ssb1_KRR-AAA migrated with ribosomes, but ribosome attachment was clearly reduced in comparison to wt Ssb1 (Fig. 2e). Thus, the KRR-region of Ssb seemingly contributes to ribosome binding but is not sufficient for this interaction since the loss of 13 C-terminal amino acids abolishes ribosome association of this Hsp70.

To provide further evidence that the KRR-motif and the C-terminus of Ssb are directly involved in targeting to the ribosome, we created a FLAG-tagged version of Ssa1 (Ssa1_RB) carrying the two ribosome binding (RB) sites of Ssb1: residues 601–13 and the KRR-motif (Supplementary Fig. 5a). As controls we designed a FLAG-tagged variant of wt Ssa1 and Ssa1ΔC (corresponding to Ssb1Δ601–13). Expression of these Ssa1 versions in *ssb1,2*Δ cells did not complement growth under the conditions tested (Supplementary Fig. 5b). However, introduction of the two Ssb ribosome-binding regions enhanced binding of Ssa1_RB to ribosomes compared with wt Ssa1 and even more

pronounced to Ssa1ΔC (Supplementary Fig. 5c; note that the Ssb-specific antibody detects the Ssb1 residues in Ssa1_RB). Although a small proportion of wt Ssa1 could be detected in the pellet fraction, Ssa1_RB showed an enhanced migration with ribosomes in the sedimentation assay, which was reduced again under higher salt conditions (Supplementary Fig. 5d). We conclude that the two Ssb ribosome-binding regions promote targeting of other Hsp70s to the translation machinery suggesting that these Ssb regions directly interact with ribosomes.

In conclusion, we identified two Ssb specific regions within the SBD and CTD of Ssb that are characterized by a basic net charge, and that are involved in the interaction with ribosomes. We suggest, that the C-terminus of Ssb represents an essential site that mediates the key contact to anchor Ssb on ribosomes while interaction via the KRR-region in the SBD is non-essential but may support the correct positioning of the domain for an efficient interaction with nascent polypeptides.

**A ribosome-binding mutant of Ssb is functional *in vivo***. Loss of ribosome binding is detrimental for nascent chain binding chaperones such as bacterial Trigger Factor and eukaryotic NAC, leading to a loss of function phenotype in both cases[31,32]. Thus, we wondered how strongly Ssb is impaired in the folding of newly synthesized proteins, as well as in ribosome biogenesis and translation when ribosome binding is abolished. To address these important questions, we investigated the Ssb1Δ601–13 mutant that completely lost its intrinsic ability to autonomously bind to ribosomes (Fig. 3a). First, we monitored the cellular localization of wt or mutant Ssb1 N-terminally fused to yEGFP in *ssb1,2Δ* cells. Both Ssb1 versions complemented growth (Fig. 3b) and showed a homogenous cytosolic distribution in the cell, and neither mislocalization nor aggregation of the constructs could be observed (Fig. 3c). Next, we performed polysome profiling to investigate whether loss of ribosome attachment of Ssb has any influence on the level of mature and translating ribosomes. Total lysates were prepared from different yeast cells and loaded on a sucrose density gradient to separate ribosomal species by centrifugation. Subsequent fractionation of the gradient reports about the 40S, 60S, 80S and polysome contents. Cells lacking Ssb show a decrease of ribosomal subunits, especially of the 60S resulting in reduced translation as evident by a reduction of 80S and polysome peaks (Fig. 3d, left;[31]). These defects were complemented by plasmid encoded Ssb1 expressed under its authentic promoter (Fig. 3d, middle). Western blot analysis of the fractions resulting from polysome profiling detected Ssb as expected in the soluble part, as well as bound to ribosomal subunits, translating monosomes and polysomes (Fig. 3d middle, bottom;[14]). Remarkably, cells expressing the ribosome-binding mutant Ssb1Δ601–13 showed a profile similar to that of wt cells with a restored 60S peak, as well as normal monosomes and polysomes (Fig. 3d, right). As anticipated, Ssb1Δ601–13 did not migrate with ribosomes and was only detectable in the soluble fractions (Fig. 3d right, bottom).

Next, we checked the expression levels of different ribosomal proteins known to be reduced in *ssb1,2Δ* cells using total lysates[31]. We found the characteristic reduction of ribosomal proteins in cells lacking Ssb (Fig. 3e,f), but expression of the mutant fully substituted for wt Ssb. Levels of functionally connected NAC and RAC were not affected by the expression of Ssb1 mutant compared with wt (Fig. 3e,f).

The absence of Ssb also causes aggregation of ribosomal and other proteins[16,31]. Therefore, we isolated the aggregated protein material from *ssb1,2Δ* cells transformed with either empty vector, wild type or mutant Ssb1. In contrast to *ssb1,2Δ* cells, which revealed strong protein aggregation (Fig. 3g), both, the expression of wt and mutant Ssb1 fully complemented the defect of this strain and showed only background aggregation comparable to wt cells.

Taken together, these data indicate that the Ssb1Δ601–13 mutant lacking the intrinsic ribosome binding ability is surprisingly fully active during ribogenesis, translation and protein folding. Thus, in contrast to other ribosome-associated chaperones, direct ribosome binding of Ssb *per se* is not a prerequisite for its *in vivo* functionality.

**Ssb versions interact with peptide substrates *in vitro***. Next, we investigated the potential differences in substrate binding of wt and mutant Ssb. To compare substrate binding of Ssb with classical Hsp70s, we used MD simulations of a crystallized DnaK–peptide complex (Fig. 4a;[36]) and a modelled Ssb1 variant (Fig. 4b). Binding of canonic substrate peptides such as NRLLLTG or APPY (ref. 37) was simulated and revealed an overall very similar interaction pattern for both complexes, suggesting that Ssb in general should be able to bind canonic Hsp70 substrate peptides.

To further address this issue we monitored the interaction of Ssb with biotinylated APPY peptide via size exclusion chromatography (Fig. 5a). We incubated APPY either alone or together with Ssb protein and separated the free peptide from Ssb-bound ones via size exclusion chromatography. Fractions upon gel filtration were spotted from high to low molecular weight followed by detection of either Ssb or peptide. APPY alone elutes in fractions no. 29–34 (Fig. 5b, left) whereas Ssb runs faster and elutes in fractions 7–12 (Fig. 5b, right). If incubated together APPY shifted into earlier Ssb-containing fractions, indicating an interaction with Ssb. A similar APPY shift could be detected for the ribosome-binding deficient Ssb1Δ601–13, indicating *in vitro* functionality of this mutant also with respect of substrate interaction (Fig. 5b). To prove that the APPY–Ssb1 interaction is specific we either pre-treated Ssb with ATP to force the open conformation (Ssb$_{ATP}$) or with apyrase that hydrolyses ATP to ADP/AMP, leading to lid closure of the Hsp70 (Ssb$_{ADP}$). After that, we added APPY peptide and in the case of Ssb$_{ATP}$ we incubated with apyrase subsequently, to stabilize the Hsp70–peptide complex. Analyses upon gel filtration showed that more APPY shifted into the Ssb containing fractions if added to Ssb in the ATP-open conformation before ATP hydrolysis (Supplementary Fig. 6a). Quantification of the shifted APPY signals showed that the peptide interaction was reduced to 71% in the case of Ssb$_{ADP}$ (Supplementary Fig. 6b). This proves that the nucleotide state of Ssb is critical for an efficient peptide binding and that the canonical Hsp70 ATPase cycle is involved in this interaction. Furthermore, incubation with a non-Hsp70 substrate peptide P2-beta, which is characterized by a hydrophobic cluster flanked by negatively charged residues, did not result in a peptide shift into Ssb-containing fractions (Supplementary Fig. 6c). We conclude that Ssb1 and Ssb1Δ601–13 bind to canonical Hsp70 peptide substrates in a specific manner similar to other Hsp70 chaperones.

We tried to determine also the affinities of wt and mutant Ssb1 for peptide substrates using fluorescence polarization measurements. Therefore, we monitored anisotropy of IAANS-labelled model peptide σ[32]-Q132-Q144-C, for which DnaK possesses a $K_d$ of 78 nM (ref. 38). In general, we could detect an increase in peptide anisotropy for wt and mutant Ssb1, without major differences between both versions suggesting a similar $K_d$ (Fig. 5c). However, it was not possible to determine the exact $K_d$ of any protein–peptide interaction since the binding curves were not reaching saturation when titrating even up to 100 μM of Ssb1. Nevertheless, it could be argued that Ssb1Δ601–13 has higher peptide association and dissociation rates resulting in a similar $K_d$. This is important, since for DnaK it is known that alterations in the lid increase the off-rate of peptides, as this region constitutes a physical lid-like barrier to substrate release[39]. If this were the case for the ribosome-binding deficient Ssb1Δ601–13 as well, one could argue that the mutant is not hampered in ribosomal interaction but rather shows a higher degree of nascent polypeptide release and is therefore lost from polysomes. Therefore, we monitored peptide release kinetics of wt and mutant Ssb and did not find any significant differences (Fig. 5d) but a similar $k_{off}$ as of DnaK (0.001 s$^{-1}$;[38]). Thus, *in vitro* peptide binding and release is similar for wt and mutant Ssb1 protein, indicating that the C-terminal deletion that abolishes ribosome binding does not influence substrate detachment, at least *in vitro*.

**Ribosome-binding deficient Ssb critically depends on RAC**. Ssb associates autonomously with ribosomes[26], but ATP hydrolysis

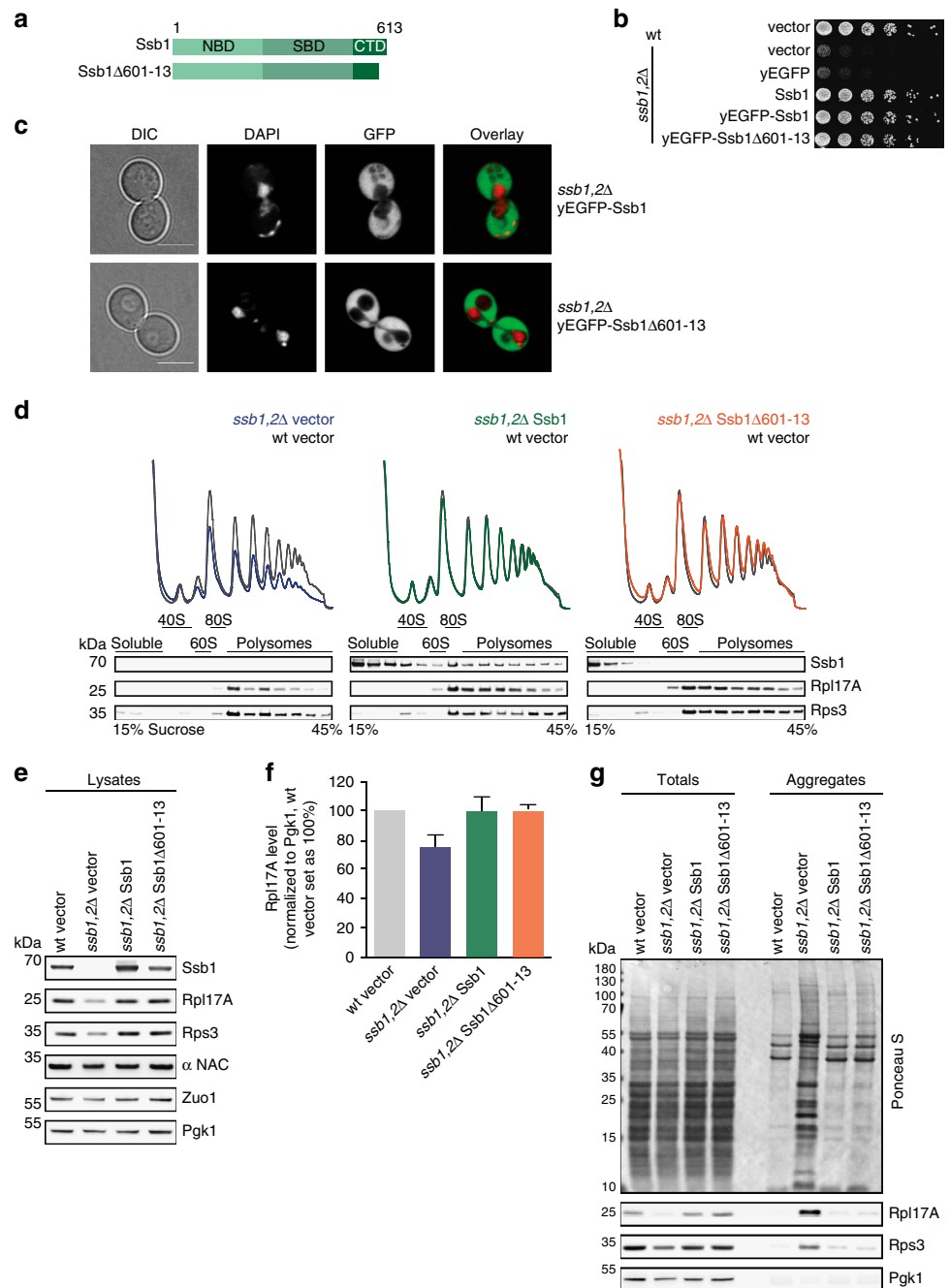

**Figure 3 | *In vivo* characterization of Ssb1Δ601-13 shows complementation of pleiotropic *ssb1,2Δ* phenotypes.** (**a**) Schematic overview of Ssb domains and constructs used. Ssb1Δ601-13 lacks 13 C-terminal residues. (**b**) Growth analysis of Ssb-GFP constructs. Wild type (wt) or *ssb1,2Δ* cells transformed with empty vector or Ssb-GFP versions were adjusted to $OD_{600} = 0.4$ and spotted in fivefold serial dilutions onto a YPD-plate, which was incubated at 30 °C for two days. (**c**) Cellular distribution of yEGFP-Ssb1 and yEGFP-Ssb1Δ601–13 analysed by confocal fluorescence microscopy. Transformed cells were grown to early exponential phase and applied onto agarose slices. Differential interference contrast (DIC) with scale bar = 5 µm; DAPI was used for nuclei staining, overlay merges DAPI and GFP. (**d**) For polysome profiling wt cells transformed with empty vector (grey) or *ssb1,2Δ* cells transformed with either empty vector (blue) or Ssb1 constructs (green, orange) were grown to early exponential phase. Lysates were adjusted and 18 $A_{260}$ units on top of a 15–45% (w/v) sucrose gradient were ultracentrifuged followed by gradient fractionation from top to bottom and $OD_{254}$ monitoring (top). Fractions were analysed via immunoblotting (bottom). Wt profiles at the background serve as control. (**e**) Protein expression of wt or *ssb1,2Δ* cells transformed with either empty vector or Ssb1 constructs. Cells were grown to early exponential phase, lysed and protein levels were adjusted followed by immunoblotting. (**f**) Quantification of Rpl17A protein levels as shown exemplarily in **e**. Rpl17A signals were normalized to Pgk1 loading control and wt cells transformed with empty vector were set as 100%. Error bars represent s.e.m. of at least three independent experiments. (**g**) Quantitative analysis of aggregates isolated from wt and *ssb1,2Δ* cells transformed with empty vector or Ssb1 constructs. Samples were blotted followed by Ponceau S staining and immunological detection.

and Ssb interaction with nascent chains is stimulated by the interaction with its ribosome-bound co-factor RAC (refs 17,18). Thus, we asked whether Ssb1Δ601–13 lacking its intrinsic ability

to interact with ribosomes still cooperates with RAC and whether this interaction is crucial for function. We used a strain lacking both RAC and Ssb (*Ssb-RAC*Δ), which was transformed with

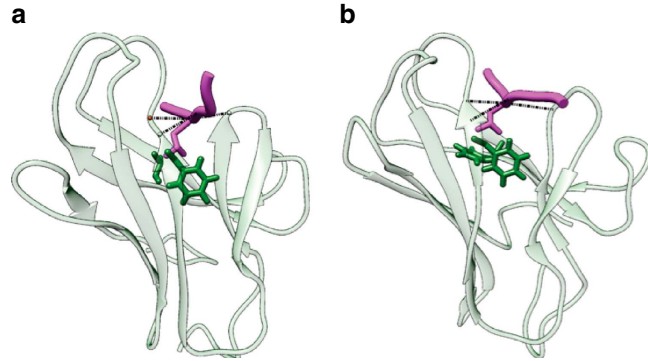

**Figure 4 | Simulated structures of DnaK and Ssb show qualitatively identical substrate binding.** (**a**) DnaK-peptide complex after 750 ns MD (molecular dynamics) simulation. SBD and peptide (NRLLLTG) are displayed in transparent green and magenta respectively. The most stable protein-peptide hydrogen bonds (indicated by dotted black lines) are S427-L3 (89%), S427-L5 (98%) and M404-L4 (86%) (with percentages in brackets representing stability during simulation). Residues forming hydrophobic contacts are depicted in stick representation involving I438, F426 and L5. (**b**) Ssb1-peptide complex as depicted in **a**. The most stable protein-peptide hydrogen bonds are T433-L3 (82%), T433-L5 (94%) and Q410-L4 (55%), residues forming hydrophobic contacts involve F444, F432 and L5.

either empty vector or plasmids encoding wt Ssb1 or Ssb1Δ601–13 (Fig. 6a). Introduction of Ssb1 into the *Ssb-RAC*Δ background mildly improved the severe growth defect of this strain under all conditions tested due to a residual activity of Ssb in absence of RAC (Fig. 6b). In contrast, the ribosome-binding mutant did not improve growth of *Ssb-RAC*Δ cells at all. This suggests that ribosome-bound RAC is essential for Ssb1Δ601–13 to display its function, whereas wt Ssb can partly operate without its ribosome-bound co-factor.

Next, we performed polysome profiling experiments that revealed that cells lacking Ssb-RAC show severe defects in ribogenesis and translation evident by a reduction of free 60S subunits and a drop of monosomal and polysomal translation (Fig. 6c, left). Transformation of *Ssb-RAC*Δ cells with wt Ssb1 restored the level of 60S subunits and improved translation, especially of monosomes and early polysomes (Fig. 6c, middle; Fig. 6d, top). Detection of Ssb1 showed that it migrates with all ribosomal fractions even in the absence of RAC (Fig. 6c, middle, bottom;[26]). In contrast, *Ssb-RAC*Δ cells transformed with mutant Ssb1Δ601–13 showed no differences in comparison to the empty vector control (Fig. 6c, right; Fig. 6d, bottom), indicating that in the absence of RAC Ssb1Δ601–13 is neither able to transiently interact with ribosomes (Fig. 6c, right bottom) nor to restore ribogenesis or translation. This is further underlined by the fact that expression levels of ribosomal proteins are reduced to similar levels in *Ssb-RAC*Δ cells transformed with either empty vector or the ribosome-binding Ssb mutant and only transformation with wt Ssb1 could improve these levels to a certain extent (Fig. 6e,f).

Finally, we analysed the interaction of wt and mutant Ssb1 (Fig. 7a) with nascent polypeptides by pulse labelling with [35]S-methionine to monitor newly synthesized proteins. Subsequently, ribosome-nascent chain complexes were isolated (Fig. 7b,e) and Ssb versions were co-immunoprecipitated from the ribosomal fraction (Fig. 7c,f). In wt cells, Ssb1 associated with nascent polypeptides as evident by the smear of labelled nascent chains that co-immunoprecipitated with Ssb1 (Fig. 7c,d). Even in the absence of RAC wt Ssb1 was able to interact with nascent polypeptides albeit with reduced efficiency. This is feasible, since RAC interacts with Ssb and is important for stimulating its

substrate binding. Strikingly, the ribosome-binding mutant Ssb1Δ601–13 completely lost the ability to bind to nascent polypeptides in the absence of RAC as evident by strongly reduced pull-down of radiolabeled nascent chains, similar to background levels (Fig. 7c,d). Conversely, in the presence of RAC the Ssb1Δ601–13 mutant was still able to associate with nascent polypeptides, albeit the efficiency is much lower compared with wt Ssb1 (Fig. 7f,g). The reduced binding of Ssb1Δ601–13 to nascent chains in the presence of RAC likely reflects the fact that this mutant is no longer able to bind directly to the translation machinery but still interacts with RAC. We conclude that autonomous ribosome binding of Ssb is crucial for its interaction with nascent polypeptides in the absence of RAC.

In summary, our data indicate that in the absence of ribosome-associated RAC Ssb needs its intrinsic ribosome-binding property to interact with the translation machinery for *in vivo* functionality. In other words, *in vivo* activity of the ribosome-binding mutant Ssb1Δ601–13 critically depends on its ribosome-associated co-chaperone RAC. This is reminiscent of the mammalian RAC system, where only mRAC is ribosome-attached and targets cytosolic Hsp70 to nascent polypeptides[19]. Taking into account that Ssb1Δ601–13 still interacts with RAC and the nascent chain but no longer with the ribosome directly, we assume that this mutant is still functional by operating in a similar manner as the mammalian Hsp70-RAC system.

**Discussion**

In this study we elucidated the molecular details underlying ribosomal interaction of the specialized Hsp70 Ssb and investigated the importance of ribosome association for the chaperoning function of Ssb *in vivo*. Our findings resolve long-standing open questions, as ribosome binding of Ssb remained enigmatic until now. Ssb shows a high degree of identity to the cytosolic Hsp70 version Ssa and lacks obvious ribosome-binding domains or elements. This is in contrast to other ribosome-associated chaperones such as Trigger Factor in bacteria or the yeast Hsp40s Zuo1 or Jjj1, which involve discrete ribosome-binding domains[40,41].

Ribosome binding of Ssb is multilayered involving direct interactions mediated by two basic regions and its contact with the ribosome-associated co-chaperone RAC. Both regions are characterized by an enrichment of positively charged amino acids. The key contact to ribosomes is mediated via 13 residues of the C-terminal lid (603–13, KRKR) and a second contact with lower affinity is provided by a KRR-motif (residues 428–30) within the SBD (Fig. 8). Deletion or substitution of positive charges in the C-terminus of Ssb showed the strongest impact on ribosomal interaction, qualifying this region as a major contact of Ssb to ribosomes. Importantly, this region is highly conserved in all Ssb homologues and thus may act as a general ribosome-docking site for this Hsp70 subfamily. The additional attachment via the KRR-region, which showed mild effects on mutation, expands this interaction and likely serves for correct positioning of the SBD in close proximity to the ribosomal tunnel exit. Both ribosome-binding regions of Ssb involve positively charged amino acids, which is in excellent agreement with earlier studies showing that Ssb, bound to non-translating ribosomes, could be stripped off by high salt treatment[11,12]. This suggests mainly electrostatic interactions between Ssb and the ribosome.

We thoroughly characterized the ribosome-binding mutant Ssb1Δ601–13 *in vitro* and *in vivo*. *In vitro* we found that wt and mutant Ssb1 similarly well bind canonical Hsp70 peptide substrates albeit we were not able to determine $K_d$ values for this interaction. One explanation for this observation might be a much lower affinity of Ssb towards peptide substrates compared

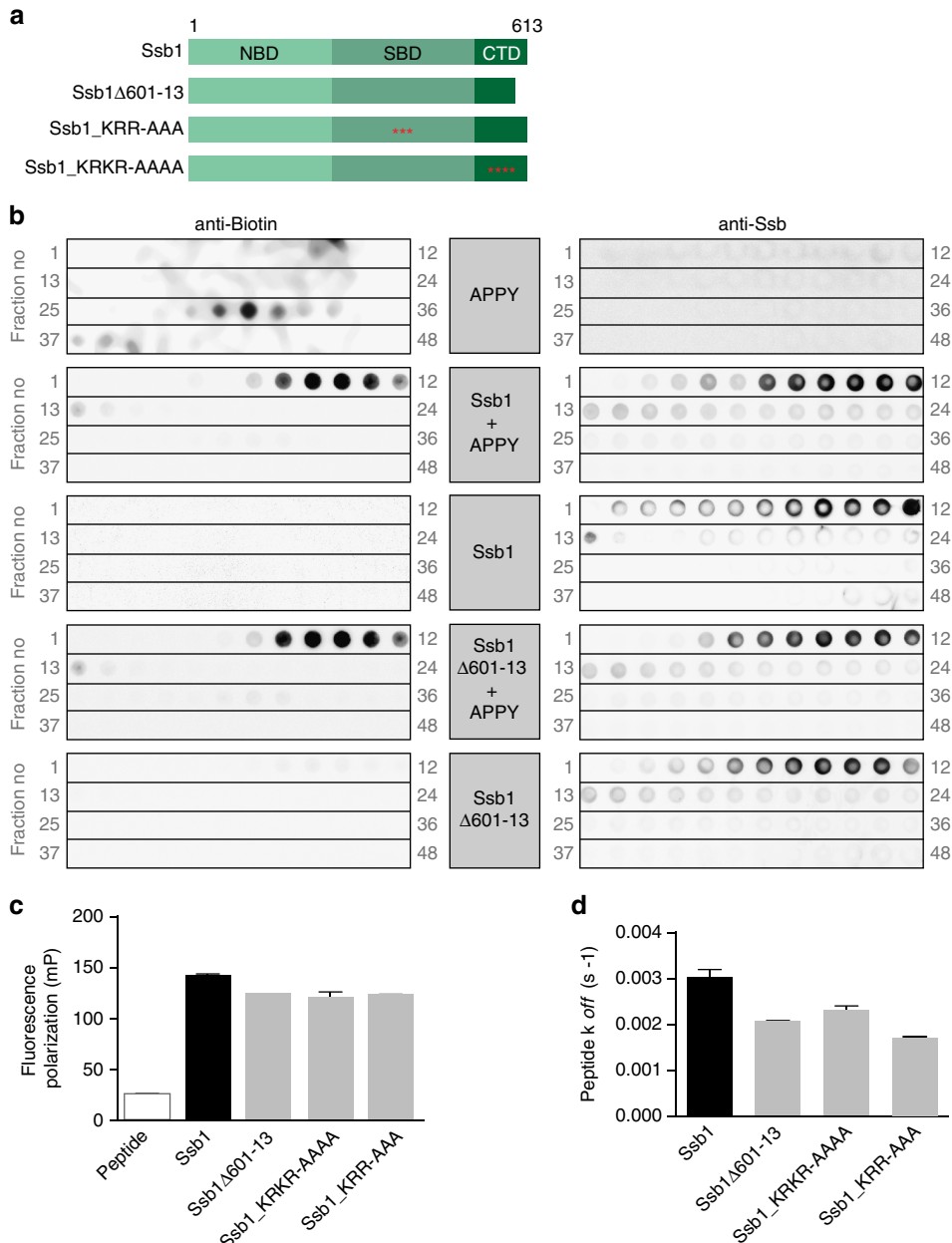

**Figure 5 | Wild type and mutant Ssb1 interact with canonic Hsp70 substrate peptides *in vitro*.** (**a**) Schematic overview of Ssb domains and the constructs used. Ssb1Δ601–13 lacks 13 C-terminal residues; red asterisks mark point mutations (Ssb1_KRR428AAA; Ssb1_KR603AA-K608A-R613A). (**b**) Analysis of wild type (wt) and mutant Ssb1 interaction with APPY-peptide analysed via gel filtration. Biotin labelled APPY peptide was incubated alone or with either wt or mutant Ssb1. Ssb1 or Ssb1Δ601–13 alone served as controls. Samples were applied to size exclusion chromatography to separate free peptide from Ssb-bound one. The resulting elutions were fractionated and spotted from high to low molecular weight (grey numbers indicate fractions) onto nitrocellulose membranes followed by immunological detection of either APPY (via HRP-StrepTactin) (left) or Ssb1 (right). (**c**) Peptide binding analyses of wt and mutant Ssb1 using fluorescence anisotropy. Wt or mutant Ssb1 was incubated with (IAANS)-labelled peptide σ32-Q132-Q144-C followed by fluorescence polarization measurements. (**d**) Peptide release kinetics of wt or mutant Ssb1 using (IAANS)-labelled peptide σ32-Q132-Q144-C. Error bars represent s.e.m. of at least three independent experiments.

with other Hsp70 chaperones. Such a scenario would exclude stable binding of Ssb to substrates in the cytosol. Alternatively, the interaction with RAC and ribosomes might be important for high affinity substrate binding of Ssb.

Surprisingly, loss of autonomous ribosome binding of Ssb did not affect its functions *in vivo* including *de novo* protein folding, ribogenesis and translation activity. This is in contrast to other ribosome-associated chaperones, as ribosome-binding deficient Trigger Factor no longer supports nascent polypeptide folding

and an analogous NAC mutant provokes growth defects and protein misfolding in *ssb1,2Δ* cells[31,32,42]. However, Trigger Factor and NAC act autonomously, while Ssb activity involves the ribosome-associated co-factor RAC. This suggests a RAC-mediated interaction of Ssb with ribosome-nascent chain complexes, even for mutant Ssb that is not able to directly bind to the translation machinery. Thus, the Ssb ribosome-binding mutant pictures the scenario of higher eukaryotes, where the mammalian mRAC recruits cytosolic Hsp70 to promote co-

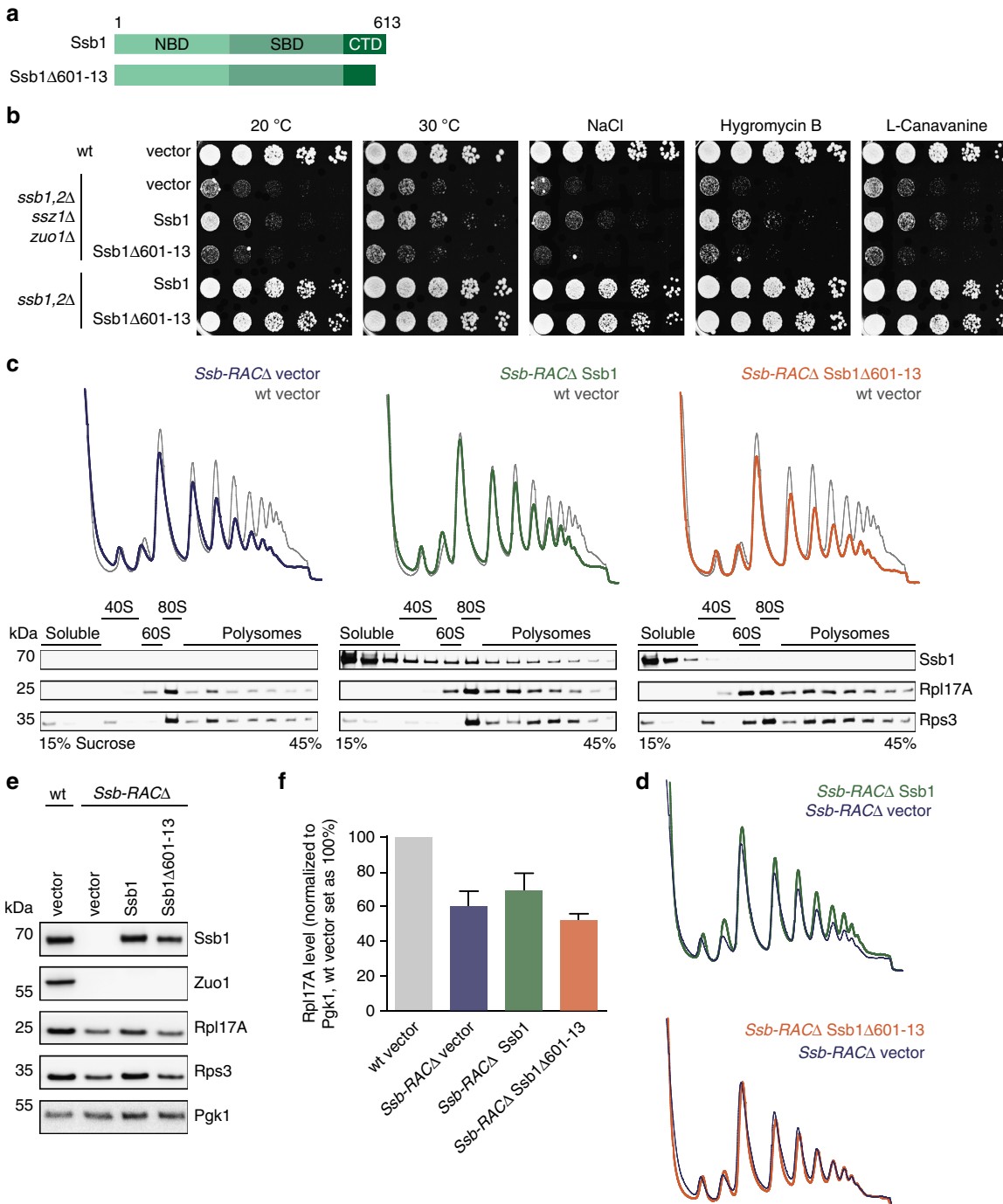

**Figure 6 | The Ssb1Δ601-13 ribosome-binding mutant needs its co-factor RAC for *in vivo* functionality.** (**a**) Schematic overview of Ssb domains and the constructs used. Ssb1Δ601-13 lacks 13 C-terminal residues. (**b**) Growth analysis of wild type (wt), *ssb1,2Δssz1Δzuo1Δ* (*Ssb-RACΔ*) and *ssb1,2Δ* cells transformed with empty vector or Ssb1 constructs. Exponentially growing cells were adjusted to $OD_{600} = 0.4$ and spotted in fivefold serial dilutions onto SC-URA plates (that may contain certain additives), which were incubated at 30 °C for two days or at 20 °C for five days. (**c**) For polysome profiling wt cells transformed with empty vector (grey) or *Ssb-RACΔ* cells transformed with either empty vector (blue) or different Ssb1 constructs (green, orange) were grown in SC-URA media to early exponential phase. Lysates were adjusted and 18 $A_{260}$ units on top of a linear 15–45% (w/v) sucrose gradient were ultracentrifuged followed by gradient fractionation from top to bottom and $OD_{254}$ monitoring (top). Fractions were analysed via immunoblotting (bottom). The wt profile (grey) at the background of each profile served as control. (**d**) Overlay of polysome profiles of *Ssb-RACΔ* cells transformed with Ssb1 (green) or Ssb1Δ601-13 (orange) with the empty vector control (blue) as shown in **c**. (**e**) Protein expression analysis of wt cells transformed with empty vector or *Ssb-RACΔ* cells transformed with either empty vector or different Ssb1 constructs. Cells were grown to early exponential growth phase in SC-URA media, harvested and lysed. Lysates were adjusted to the same protein concentration and analysed via SDS–PAGE and immunoblotting. (**f**) Quantification of Rpl17A protein levels as shown exemplarily in **e**. Rpl17A signals were normalized to Pgk1 loading control and wt cells transformed with empty vector were set as 100%. Error bars represent s.e.m. of at least three independent experiments.

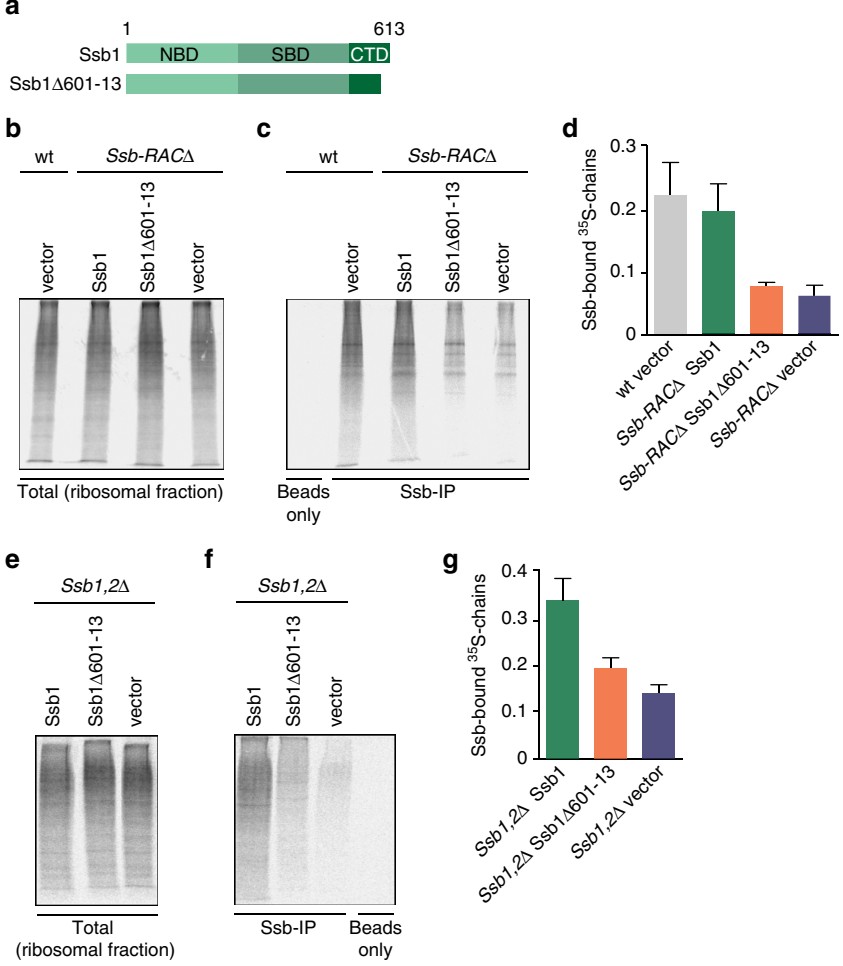

**Figure 7 | A ribosome-binding mutant of Ssb critically depends on RAC for efficient nascent chain interaction. (a)** Schematic overview of Ssb domains and the constructs used. Ssb1Δ601–13 lacks 13 C-terminal residues. Wild type (wt), *ssb1,2Δ* or *ssb1,2Δssz1Δzuo1Δ* (*Ssb-RACΔ*) cells were transformed with either empty vector, wt Ssb1 or Ssb1Δ601–13. Cells were grown to $OD_{600} = 0.5$, starved for 30 min in -Met medium and pulsed with $^{35}$S-Met for 1.5 min. Cells were harvested, lysed and ribosomal pellets were isolated using a 25% (w/v) sucrose cushion. Ssb immunoprecipitations (IPs) were performed using Ssb antibody crosslinked to ProteinG dynabeads. **(b)** Autoradiogram of $^{35}$S-labelled nascent chains from ribosomal pellets of transformed wt or *Ssb-RACΔ* cells. **(c)** Immunoprecipitation of Ssb-associated RNCs (ribosome nascent-chain complexes) from ribosomal pellets of transformed wt or *Ssb-RACΔ* cells. **(d)** Quantification of Ssb-associated RNCs from ribosomal pellets of the different strains as shown in **c** compared with corresponding ribosomal pellets as shown in **b**. **(e)** Autoradiogram of $^{35}$S-labelled nascent chains from ribosomal pellets of transformed *Ssb1,2Δ* cells. **(f)** Immunoprecipitation of Ssb-associated RNCs from ribosomal pellets of transformed *Ssb1,2Δ* cells. **(g)** Quantification of Ssb-associated RNCs from ribosomal pellets of the different strains as shown in **f** compared with corresponding ribosomal pellets as shown in **e**. Error bars represent s.e.m. of at least three independent experiments.

translational protein folding[19]. This assumption is strongly supported by our finding that Ssb1Δ601–13 is no longer functional in the absence of RAC, while wt Ssb maintains residual activity. Furthermore, overexpression of the mammalian Zuotin homologue Mpp11 complements the phenotypes of *zuo1Δ* cells but acts together with cytosolic Ssa, not with Ssb (refs 9,19,20). This suggests that also in the yeast *S. cerevisiae* ribosome association of RAC is sufficient to target cytosolic Hsp70 to the ribosome. The residual functionality of Ssb observed in the absence of RAC might be due to two reasons, which are not mutually exclusive: First, Ssb possesses a comparably high intrinsic ATPase activity in comparison to other Hsp70s (ref. 43), which could be even higher at the ribosome, that might allow efficient nascent chain binding also without the additional stimulation by RAC. A second explanation could be that Ssb is able to cooperate with other Hsp40 co-chaperones. Although earlier experiments did not detect *in vitro* ATPase stimulation of Ssb by other yeast Hsp40s like Ydj1 or Sis1 (ref. 43), it could be

shown that overexpression of the ribosome-associated Hsp40 Jjj1 partially substitutes for Zuo1 in a *RACΔ* strain[28,44]. This suggests a possible cooperation of the Hsp70 Ssb with other ribosome-attached or cytosolic Hsp40 co-chaperones in the absence of RAC.

Still, the question remains why ribosomal interaction of Ssb is evolutionarily conserved in fungi, albeit clearly not essential. We can only speculate that for unicellular yeast with a high translation rate and fast cell division it might be beneficial to optimize its co-translational protein folding capacity by tethering an Hsp70 directly to the ribosome. Furthermore, in contrast to Ssb, RAC is substoichiometric to ribosomes[12], demanding a RAC-independent mode of ribosome binding for Ssb. We propose that the multivalent ribosomal contacts via the CTD and SBD of Ssb in combination with its interaction with ribosome-tethered RAC position Ssb in an optimal orientation to the ribosomal tunnel exit to guarantee an efficient interaction with the nascent chain (Fig. 8).

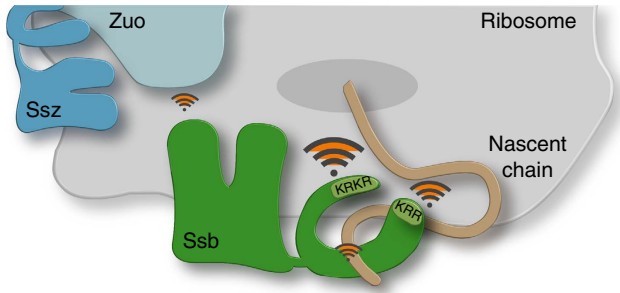

**Figure 8 | Model of the multifaceted interactions of the yeast Hsp70 Ssb at the ribosomal exit site.** Ssb possesses two intrinsic ribosome-binding sites: basic amino acid side chains within residues 603–13 of the C-terminal domain (KRKR) mediate a strong key contact to the ribosome, KRR (residues 428–30) of the substrate-binding domain represents a second attachment point which might serve for correct positioning of Ssb at the ribosomal exit site. Further indirect ribosomal contacts of Ssb are mediated via binding of the nascent polypeptide and by transient J-domain interaction with Zuotin (Zuo) that forms the ribosome-associated complex RAC together with the Hsp70 Ssz.

On the basis of findings of this study, further important questions can be addressed: (i) which are the contact sites of Ssb at the ribosome? Ssb likely binds in close proximity to the co-factor RAC and the nascent chain; (ii) how does Ssb detach from the ribosome upon nascent chain interaction? It is tempting to speculate that substrate binding and ATP hydrolysis induce conformational changes of Ssb that promote ribosome detachment for example by movement of the lid towards the substrate binding pocket. A similar mechanism is described for Hsp70s of cellular organelles like the endoplasmic reticulum or mitochondria. In both cases Hsp70s interact with the translocation machinery and detach from these complexes upon nascent polypeptide interaction and in dependency of their specific co-factors[45,46]. We suggest a similar mechanism leading to ribosome detachment of Ssb upon nascent polypeptide interaction and RAC-stimulated ATP hydrolysis.

## Methods

**Strains and plasmids and growth conditions.** All yeast strains used in this study are derivatives of the BY4741 strain, except of $ssb1,2\Delta ssz1\Delta zuo1\Delta$, which is a DS10 derivative, and are described elsewhere[31].

The pRS316 vector[47] containing Ssb-promoter and –terminator regions was used for all transformation experiments. Insertion of wt-Ssb1 ORF or variations thereof were created via Fusion-PCR. All Ssb1 mutations and GFP-fusions were done according to standard mutagenesis protocols. Ssa1 constructs were cloned into the same pRS316 vector as Ssb versions including N-terminal Flag-tagging. Ssa1ΔC lacks residues 593–642. Ssa1_RB lacks residues 593–642, KSE421 is substituted by the corresponding Ssb residues KRR and the Ssb1 residues 601–13 are fused to the C-terminus.

Unless described otherwise cells were grown at 30 °C in YPD (1% (w/v) Bacto Yeast Extract, 2% (w/v) Bacto-Peptone, 2% (w/v) Dextrose) or defined synthetic complete (SC) media (6,7 g l$^{-1}$ Bacto-Yeast Nitrogen Base w/o amino acids, 2 g l$^{-1}$ amino acid mix, 2% (w/v) dextrose). Spotting assays were performed by adjusting OD$_{600}$ of exponentially growing cultures to 0.4, followed by spotting of fivefold serial dilutions onto selection plates, which were incubated for 2 days at 30 °C as indicated or 5 days at 20 °C. Plates supplemented with additives contained 0.8 M NaCl, 25 μg ml$^{-1}$ Hygromycin B or 0.75 μg ml$^{-1}$ L-Canavanine respectively and were incubated for 2 days at 30 °C.

**Antibodies and Western blot analyses.** Protein samples were analysed according to standard protocols via SDS–PAGE followed by Coomassie staining or electro blotting onto nitrocellulose membrane (GE Healthcare). First and secondary antibodies were used 1:10,000 diluted. Polyclonal antibodies against Ssb1, NAC, Zuo1, Rpl17A and Rps3 are already described elsewhere[31]. Pgk1 (Novex, Life Technologies, 459250) and HRP-coupled antibodies are commercially available (Dianova, 715-035-151, 711-035-152), as well as HRP-coupled StrepTactin (IBA, 2-1503-001). Uncropped images of the most important blots can be found as Supplementary Fig. 7.

**Modelling of the Ssb1 structure.** The structure model for Ssb1 was generated by the software modeller (version 9.13; ref. 48) using an alignment of the Ssb1 sequence to sequence of the Hsp70 DnaK chaperone with known structure (PDB ID: 2kho;[49]). The homology-modelled structure was then imported into PyMol (version 1.7.7.2; The PyMOL Molecular Graphics System, Schrödinger, LLC) for colouring and highlighting of the relevant sections and residues.

**Ssb1 simulations.** Simulations were performed using Gromacs-4.6.5 package[50], Gromos54a7 force-field[51] and the SPC/E water model[52]. Simulations were performed on the respective apo forms with SBD and CTD (setup-I) or the SBD with peptides bound (setup-II). Starting structures were taken from the PDB (ref. 53) if available or modelled. After energy minimization (and 4–7 ns equilibration with and w/o position restraints) simulations were carried out for 750 ns at NPT conditions with a 2 fs time step (298 K and 1 bar) using a v-rescale thermostat[54] and a Berendsen barostat[55]. Electrostatic interactions were evaluated using the PME method[56] and VDW interactions were truncated at 1.2 nm. For validation, simulations have been repeated three times with slightly varying starting conditions yielding qualitatively identical results.

**Ribosome sedimentation assay.** Ribosome sedimentation was adapted from[57]. Cells were grown to OD$_{600}$ = 0.6–0.8, harvested, resuspended in lysis buffer (20 mM HEPES KOH pH 7.4, 50 mM KAc, 2 mM MgAc, 2 mM DTT, protease inhibitors, 1 mM PMSF that may contain 1 mM puromycin) and lysed via glass bead disruption. Lysates were cleared twice (14,000g, 10 min, 4 °C) and adjusted to 4.5 A$_{260}$ units. 1.5 A$_{260}$ units were TCA-precipitated as total sample according to standard protocols. Another 1.5 A$_{260}$ units were loaded onto 3-fold volume of a 25% (w/v) sucrose cushion prepared with lysis buffer and centrifuged for 90 min at 200,000g (S140-rotor, Sorvall) at 4 °C. The supernatant was TCA-precipitated according to the total sample and the resulting pellets were resuspended in the same volume as the ribosomal pellet after ultracentrifugation. Total, supernatant and pellet fractions were analysed via SDS–polyacrylamide gel electrophoresis (SDS–PAGE) and Western blotting.

**Fluorescence microscopy.** Cells were grown to early exponential phase (OD$_{600}$ = 0.5) in SC-URA media (pH 5.6) and shifted for DNA-staining to media with a pH = 7.0. DAPI was added to a final concentration of 10 μg ml$^{-1}$, cells were incubated for 30 min at 30 °C and finally immobilized on thin-coated agarose slides (1% (w/v) in SC-URA). Fluorescence microscopy was performed using a confocal Leica TCS SP8 microscope equipped with a 63x oil immersion objective lens (NA 1.4). Digital images were processed using Fiji software.

**Polysome profiling.** Polysome profiles were performed as described elsewhere[31] with the following adaptations. Yeast cells were grown at 30 °C to an OD$_{600}$ = 0.6–0.8, quickly harvested using a vacuum pump and flash-frozen in liquid N$_2$. Frozen pellets were powdered together with 1 ml frozen lysis buffer (20 mM HEPES KOH pH 7.4, 100 mM KAc, 2 mM MgAc, 0.5 mM DTT, 1 mM PMSF, 1x TmComplete protease inhibitor cocktail (Roche), 100 μg ml$^{-1}$ cycloheximide) using a pre-cooled Retsch Mill MM400 and applying 22 Hz for 30 s. The powder was thawed, cleared by centrifugation at 16,000g for 10 min at 4 °C, and adjusted to same A$_{260}$ units. 18 A$_{260}$ units were loaded onto 11 ml of a linear 15–45% (w/v) sucrose gradient, prepared in lysis buffer w/o PMSF (Gradient Master; Biocomp Instruments). Gradients were centrifuged in a TH-641 rotor (Sorvall) at 39,000 r.p.m. for 2.5 h at 4 °C, followed by fractionation from top to bottom with a gradient fractionator (Teledyne Isco, Inc.). A$_{254}$ signals were recorded and absorbance data were processed with PeakTrak V1.1 (Teledyne Isco, Inc.). Fractions of 500 μl were collected and analysed via SDS–PAGE and Western blotting.

**Isolation of protein aggregates.** Aggregates were prepared as described elsewhere[31] with the following adaptations. Pre-cultures, grown in SC-URA media were used to inoculate YPD media with an OD$_{600}$ = 0.1 and cultures were grown at 30 °C to OD$_{600}$ = 0.6–0.8. Cells were harvested in the presence of 15 mM sodium azide and flash-frozen in liquid N$_2$. Pellets were resuspended in 1 ml buffer I (20 mM potassium phosphate pH 6.8, 10 mM DTT, 1 mM EDTA, 0.1% Tween 20, protease inhibitors, 1 mM PMSF and 1.25 U ml$^{-1}$ DNase (Sigma) containing 3 mg ml$^{-1}$ Zymolyase-T20 (MP Biomedicals)), incubated for 15 min at RT and chilled on ice for 5 min. After sonication (Branson tip-sonifier; 8x level 4, 50% duty cycle) samples were centrifuged for 20 min at 200g and the protein concentration of the supernatants was adjusted. A sample of each normalized lysate was taken as an input control (total). Protein aggregates were sedimented at 16,000g for 20 min at 4 °C. Pellets were washed twice with buffer II (20 mM potassium phosphate pH 6.8, protease inhibitors) containing 2% (v/v) Nonidet P-40 (NP40), and once with buffer II w/o NP40, sonicated (6x level 4, 50% duty cycle), and centrifuged at 16,000g for 20 min at 4 °C. Finally, protein aggregates were resuspended in sample buffer and together with totals analysed via SDS–PAGE followed by Ponceau S staining and Western blotting.

**Purification of proteins and *in vitro* peptide interaction.** For recombinant protein expression and purification *E. coli* BL21(DE3)*/pRARE cells were transformed with a His$_6$-Smt3-Ssb1 construct (or mutations thereof). Cells were grown at 30 °C and protein expression was induced with 1 mM IPTG for 4 h after reaching $OD_{600} = 0.6$. Cells were harvested, resuspended in lysis buffer (30 mM HEPES KOH pH 7.4, 500 mM KAc, 5 mM MgCl$_2$, 10% glycerol (v/v), 1 mM β-mercaptoethanol, 5 mM ATP, 1 mM PMSF, protease inhibitors, DNase) and lysed by French press. The lysate was cleared by centrifugation for 30 min at 16,000$g$ at 4 °C, the supernatant was applied to a Ni-IDA matrix (Protino; Macherey-Nagel) and incubated for 30 min at 4 °C. After several high salt (30 mM HEPES KOH pH 7.4, 1 M KAc, 5 mM MgCl$_2$, 10% (v/v) glycerol, 1 mM β-mercaptoethanol, 5 mM ATP) and low salt (like high salt with 50 mM KAc) washes, His$_6$-Smt3-Ssb1 was eluted in elution buffer (like low salt plus 350 mM Imidazole) and dialyzed over night against low salt buffer together with 5 μg Ulp1 protease per mg protein for proteolytic cleavage of the His$_6$-Smt3 tag. A final purification step via ion exchange using a 6 ml Resource Q (Anion exchange) column (GE Healthcare) was used to remove the His$_6$-Smt3 tag, Ulp1 and potential contaminations. Fractions containing Ssb1 were pooled, dialyzed against Ssb-buffer (30 mM HEPES KOH pH 7.4, 50 mM KAc, 5 mM MgCl$_2$, 1 mM β-mercaptoethanol), frozen in liquid nitrogen and stored at − 80 °C.

Ssb-peptide binding assay based on gel filtration approach: 2 μM of biotinylated peptide (APPY: SALLQSRLLLSAPRRAAATARYC; P2-beta: GSLEEIIAEGQKC) were incubated for 30 min at RT with 4 μM purified Ssb1 protein in Ssb buffer. Afterwards, apyrase (NEB) was added for 1 h to hydrolyse ATP to ADP/AMP. For specificity analysis Ssb proteins were pre-incubated with Apyrase or ATP before adding them to the biotinylated peptide. After incubation the samples were injected onto a Superose 12 column (Pharmacia, HR10/30, 24 ml) equilibrated with Ssb buffer for size exclusion chromatography. Fractions of 0.3 ml were collected and spotted via DotBlot on a nitrocellulose membrane (GE Healthcare) followed by detection of either Ssb protein (specifically with antibodies or unspecifically via Ponceau S staining) or the biotinylated peptide (via StrepTactin-HRP).

Peptide substrate binding by Ssb using fluorescence anisotropy: 10 μM of Ssb1 wt or mutant were mixed with 1 μM of 2-(4'-(iodoacetamido)anilino) naphthalene-6-sulfonic acid (IAANS) labelled peptide σ$^{32}$-Q132-Q144-C in HKM buffer (25 mM HEPES, 150 mM KCl, 5 mM MgCl$_2$, 1 mM DTT, 10% (v/v) glycerol) and incubated at 30 °C for 1.5 h. Subsequently fluorescence anisotropy measurements (excitation filter 360–10, emission filter 450–10, dichroic filter LP410) were performed with a CLARIOstar microplate reader (BMG Labtech) using 384-well black flat bottom microplates (Corning) and a final sample volume of 30 μl.

Peptide release measurements: Dissociation rates for peptide substrate were determined by pre-incubating 1 μM Ssb1 wt or mutant protein with 1 μM of 2-(4'-(iodoacetamido)anilino) naphthalene-6-sulfonic acid (IAANS) labelled peptide σ$^{32}$-Q132-Q144-C for at least 1.5 h at 30 °C in HKM buffer before mixing with a 100-fold excess of unlabelled σ$^{32}$-Q132-Q144-C peptide in a CLARIOstar microplate reader (BMG Labtech). Changes of fluorescence intensity (excitation filter 360–20, emission filter 450–10) were measured for 1 h in 384-well black flat bottom microplates (Corning). The obtained curves were fitted by a single-exponential decay function in Prism 5.0 (GraphPad).

**Purification of ribosomes and *in vitro* ribosome binding.** Yeast ribosomes were purified as already described[31]. Yeast cells were grown in YPD media to $OD_{600} = 0.8$ at 30 °C, harvested and resuspended in CSB (300 mM sorbitol, 20 mM HEPES KOH pH 7.4, 1 mM EGTA, 5 mM MgCl$_2$, 10 mM KCl, 10% (v/v) glycerol) supplemented with 1 mM PMSF, 5 mM ATP, 1x TmComplete protease inhibitor cocktail (Roche), 5 μg ml$^{-1}$ puromycin and lysed by French press. The lysate was cleared by centrifugation at 16,000$g$ for 30 min at 4 °C and layered on top of two volumes of a 20% (w/v) sucrose cushion in CSB without sorbitol containing 1 M KAc. Ribosomes were sedimented for 4 h at 200,000$g$, resuspended in 4 ml CSB and sucrose cushion centrifugation was repeated once. The ribosomal pellet was finally resuspended in CSB and shock-frozen in liquid N$_2$.

Yeast ribosomes and different Ssb proteins were thawed and cleared from potential aggregates by centrifugation at 16,000$g$ for 20 min at 4 °C. Ribosomes (0.5 μM) were incubated with 0.5–4 μM protein in Ssb-buffer for 30 min at 30 °C and layered onto a 30% (w/v) sucrose cushion prepared with Ssb-buffer. Samples were centrifuged for 70 min at 220,000$g$ at 4 °C, the supernatant was TCA-precipitated and the resulting pellet was resuspended in sample buffer. The ribosomal pellet was resuspended in the same volume of sample buffer and supernatant and pellet fractions were analysed via SDS–PAGE followed by Coomassie staining or immunoblotting.

**Nascent chain labelling and Ssb immunoprecipitation.** Wild type, *Ssb1,2Δ* or *Ssb-RACΔ* (*Ssb1,2ΔSsz1ΔZuo1Δ*) cells were transformed with either empty vector or plasmids encoding wt Ssb1 or Ssb1Δ601–13. Cells were grown in YPD to $OD_{600} = 0.5$ and then starved for 30 min in -Methionine medium, before pulsing with 100 μCI ml$^{-1}$ $^{35}$S-Metionine for 1.5 min. Cells were harvested on ice with 250 mM sodium azide and 0.5 mg ml$^{-1}$ cycloheximide. Cell pellets were lysed and ribosomal pellets were isolated using a 25% (w/v) sucrose cushion. Ssb IPs were performed using an Ssb antibody crosslinked to ProteinG dynabeads; negative controls were performed with wt lysates and pre-immune beads. IPs were performed at 4 °C for 2 h before washing, elution with SDS buffer and samples were separated by 15% SDS–PAGE, dried and quantified by autoradiography.

**Data availability.** The data that support the findings of this study are available from the corresponding author on request.

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

## Acknowledgements

We thank Valentin Schoop, Martin Gamerdinger and Steffen Preissler for valuable support and discussion of the manuscript and Christina Schlatterer for proofreading. This work was supported by fellowships of the Konstanz Research School Chemical Biology to M.A.H., by research grants from the German Science Foundation (DFG; SFB969) to E.D. and C.P., from NIH to J.F. and from Human Frontier in Science to E.D. and J.F. and by a guest-professorship for J.F. A.K. acknowledges postdoctoral support from NIH. Computational work was performed on the bwUniCluster and ForHLR Phase I funded within the framework program bwHPC by the State Baden-Württemberg and the DFG.

## Author contributions

M.A.H. and E.D. designed the experiments, analysed the data and wrote the manuscript. M.A.H. performed all *in vivo* experiments, cloning, protein purification, ribosome-binding and gelfiltration analyses. R.K. and M.P.M. performed fluorescence polarization measurements. S.J.F. performed fluorescence microscopy. A.J. and C.P. performed M.D. simulations. A.K., V.A. and J.F. performed nascent chain labelling experiments. T.F. performed alignments and Ssb modelling. All authors commented on the manuscript.

## Additional information

**Competing financial interests**: The authors declare no competing financial interests.

**How to cite this article**: Hanebuth, M. A. *et al.* Multivalent contacts of the Hsp70 Ssb contribute to its architecture on ribosomes and nascent chain interaction. *Nat. Commun.* **7,** 13695 doi: 10.1038/ncomms13695 (2016).

**Publisher's note**: 

