## [Peer Review File · Nature Communications]

PEER REVIEW FILE

Reviewers' comments:

Reviewer #1 (Remarks to the Author):

This manuscript addresses the importance of separate functions of Ssb in ribosome interaction, nascent chain interaction, and functional interaction with RAC. The authors identify mutations, including Ssb1Δ601-13, which support in vivo functions of Ssb functions despite the inability to stably interact with the ribosome. The authors further show that Ssb and the Ssb1Δ601-13 mutant had similar interactions with the APPY peptide. This shows that the inability of the mutant Ssb to bind to ribosomes is not due to the inability to bind nascent chains. Finally, the authors examined how loss of RAC affects the function of Ssb1Δ601-13. They show that WT Ssb1 partially rescues growth defect and polysome profile defects of *ssb*-RAC delta cells, but Ssb1Δ601-13 does not. The authors conclude that RAC plays a role in the correct positioning of Ssb near the exit tunnel.

The manuscript significantly advances and clarifies the knowledge about the mechanism by which Ssb interacts with the ribosome. The background is presented well, the data is nicely presented and rigorously tested and supports the authors' conclusions. The authors also include a useful discussion of the parallels with the analogous system in mammalian cells.

Suggested improvements

1. The authors show that binding of the APPY peptide to Ssb was reduced in the presence of ATP versus ADP (Figure S5B). How does this information fit with what is known about the role of RAC in stimulating the ATPase activity of Ssb and the interaction of Ssb with nascent polypeptides?
2. Have the authors examined whether RAC is able to stimulate the ATPase activity of Ssb1Δ601-13?

3. For clarification purposes, it would be helpful to note whether RAC is present in the experiments where purified Ssb is added to salt-stripped and puromycin treated ribosomes (Figure S2).
4. Figure 6E showing levels of ribosomal proteins in the Ssb-RAC delta strain complemented with WT Ssb or the Ssb mutant would be more convincing if quantification of the protein levels was provided.
5. Figure 7C shows that, in the absence of RAC, the Ssb1 Δ 601-13 mutant does not crosslink to nascent chains. The authors claim this points to a critical role for RAC in directing Ssb to the nascent chains. It would be helpful to also show that Ssb1 Δ 601-13 is able to bind nascent chains in otherwise WT cells. This would provide additional evidence that the primary defect of Ssb1 Δ 601-13 is ribosome interaction rather than nascent chain interaction.

Reviewer #2 (Remarks to the Author):

Many fungi harbor a special form of the molecular chaperone Hsp70, Ssb, which is associated with the ribosome. Together with the ribosome-associated J-protein complex RAC, Ssb supports de novo protein folding at the ribosome. Furthermore Ssb is involved in ribosome biogenesis. The present manuscript investigates the molecular basis for the interaction of Ssb with ribosomes.

The authors identify two regions enriched in basic residues that are conserved in Ssb sequences but not in general cytosolic Hsp70s. These regions are located in the substrate binding domain (KKR 428-430) and close to the C-terminus (K603, R604, K608 and R613). Mutational analysis showed that the latter is essential for constitutive interaction with the ribosome, while the former improves ribosome binding. However, both the 601-613 deletion and KRKR-AAAA mutants appear to be functional as long as RAC is present, suggesting that RAC transiently recruits cytosolic Hsp70 isoforms Ssb and Ssa to the ribosomal exit tunnel. In absence of RAC, wt Ssb interacts with nascent chain whereas the mutant does not. This suggests that Ssb can also cooperate with other J-domain proteins. This aspect could be investigated further.

Somewhat unsurprisingly, interaction studies with a model substrate peptide showed no substantial differences of the Ssb mutant proteins compared to wildtype.

A limitation of the manuscript is that the binding site of Ssb on the ribosome was not addressed. Nevertheless, this is an interesting, well-written story that provides additional insight into co-translational folding.

Point by point responses:

Reviewer #1

This reviewer stated that „the manuscript significantly advances and clarifies the knowledge about the mechanism by which Ssb interacts with the ribosome“, but suggested five points for improvement:

1. *The authors show that binding of the APPY peptide to Ssb was reduced in the presence of ATP versus ADP (Fig. S5B). How does this information fit with what is known about the role of RAC in stimulating the ATPase activity of Ssb and the interaction of Ssb with nascent polypeptides?*

Authors: In general Hsp70s have a lower affinity to substrates in the ATP-bound state than in the ADP-bound one (for review see e.g. Mayer, 2013). This lower affinity is due to the fact that ATP binding increases both association and dissociation rates. In their allosteric cycle Hsp70s bind substrates in the ATP-induced open conformation. Substrate binding in synergism with a J-domain protein, which is RAC in the case of Ssb, trigger ATP hydrolysis to ADP, which leads to the closed conformation of Ssb and consequently tight peptide binding (see De Los Rios & Barducci, 2014). We analyzed the specificity of the Ssb-APPY interaction by adding apyrase or ATP, which both influence the accessibility of the substrate-binding pocket. Pretreatment with apyrase led to the expected reduction of peptide binding to Ssb, as the Hsp70 changed into its ADP-closed conformation with very low substrate association rates. In contrast, addition of ATP to Ssb led to the open conformation allowing efficient peptide interaction. ATP hydrolysis then induced lid closure and tight substrate binding. During the gel filtration ATP was absent allowing us to retain more peptide bound to Ssb when ATP was present than when apyrase was added previous to incubation with the peptide.

We did not add RAC in this experiment as *in vitro* Hsp70 peptide-binding occurs without any co-factor albeit less efficient. However, published data show that RAC stimulates the ATPase of Ssb (Huang *et al.*, 2005, see also next point) as well as binding of nascent chains (Gautschi *et al.*, 2002; Willmund *et al.*, 2013) and in general ATP-hydrolysis is stimulated most efficiently by a combination of both, substrate and Hsp40 co-chaperone (Mayer & Kityk, 2015).

(Please note: Due to changes in the current manuscript the figure mentioned above is now labeled as Supplementary Fig. 6b).

2. *Have the authors examined whether RAC is able to stimulate the ATPase activity of Ssb Δ 601-13?*

Authors: We tested RAC stimulation for wt Ssb1 and Ssb1 Δ 601-13 (Ssb1 Δ C) *in vitro* using purified proteins and found that RAC stimulates the ATPase activity of both wild type and mutant Ssb protein.

Much more effort has to be put into carefully elucidating the ATPase cycle of Ssb on ribosomes driven by substrates and co-chaperones. These analyses will be complex and time consuming and are thus beyond the scope of this study. As we consider the data as too premature in its current state, we would prefer not to add it to this manuscript.

[Redacted]

3. For clarification purpose, it would be helpful to note whether RAC is present in the experiments where purified Ssb is added to salt-stripped and puromycin treated ribosomes (Fig. S2).

Authors: The purified ribosomes that we used for the *in vitro* binding analyses of wt and mutant Ssb protein had been salt-stripped and puromycin treated to generate a homogeneous population of ribosomes. Thus, these ribosomes almost completely lack associated factors like NAC, RAC or Ssb. We did not include additional RAC to these binding analyses as Ssb interacts with ribosomes independently of RAC (Rakwalska *et al.*, 2004).

We now include data in the manuscript showing the Western blot analysis of NAC and RAC in our purified ribosome fraction (see Supplementary Fig. 2b) and changed the corresponding passage of the text: "To test ribosome binding *in vitro*, recombinantly purified wt Ssb1 and Ssb1 Δ 601-13 protein was incubated with puromycin stripped and high salt washed yeast ribosomes (Supplementary Fig. 2a) that were almost completely devoid of exit site associated factors like NAC or RAC (Supplementary Fig. 2b)." Furthermore the figure legend was adapted as follows: "**b**) Salt-stripped and puromycin-treated yeast ribosomes were tested for the presence of ribosome associated factors by either SDS-PAGE and Coomassie staining (top) or by immunological detection of different proteins (bottom)."

4. Figure 6E showing levels of ribosomal proteins in the Ssb-RAC delta strain complemented with WT Ssb or the Ssb mutant would be more convincing if quantification of the protein levels was provided.

Authors: As suggested by the reviewer, we quantified the experimental data and show the statistics in Fig. 6f. This analysis clearly shows that in five independently performed biological replicates of Ssb-RAC Δ cells transformed with Ssb1 Δ 601-13 the level of Rpl17A is strongly reduced in comparison to wt cells and also lower than in Ssb-RAC Δ cells transformed with wt Ssb1.

For the sake of completeness we included this type of Rpl17A quantification also into the corresponding analysis in Fig. 3 (see Fig. 3f). This quantification clearly shows that in the presence of RAC the level of ribosomal proteins in *ssb1,2* Δ cells transformed with either Ssb1 or Ssb1 Δ 601-13 is completely restored to wt levels in both cases.

We adapted the figure legends of both figures as follows: "**f**) Quantification of Rpl17A protein levels as shown exemplarily in e). Rpl17A signals were normalized to Pgk1 loading control and wt cells transformed with empty vector were set as 100 %. Error bars represent SEM."

5. Figure 7C shows that, in the absence of RAC, the Ssb Δ 601-13 mutant does not crosslink to nascent chains. The authors claim this points to a critical role for RAC in directing Ssb to the nascent chains. It would be helpful to also show that Ssb Δ 601-13 is able to bind nascent chains in otherwise WT cells. This would provide additional evidence that the primary defect of Ssb Δ 601-13 is ribosome interaction rather than nascent chain interaction.

Authors: As suggested by the referee we performed this experiment and included the new data in the manuscript. We can show that Ssb1 Δ 601-13 is able to interact with nascent chains in the presence but not in the absence of RAC albeit less efficient than wt Ssb1. Taken together these findings support that autonomous binding of Ssb to ribosomes enhances its ability to interact with nascent polypeptides. According to that we changed the corresponding passage also in the manuscript: "Conversely, in the presence of RAC the Ssb1 Δ 601-13 mutant was still able to associate with nascent polypeptides, albeit the efficiency is much lower compared to wt Ssb1 (Fig. 7f, g). The reduced binding of Ssb1 Δ 601-13 to nascent chains in the presence of RAC likely reflects the fact that this mutant is no longer able to bind directly to the translation machinery but still interacts with RAC. We conclude that autonomous ribosome binding of Ssb is crucial for its interaction with nascent polypeptides in the absence of RAC."

Reviewer #2

Referee 2 stated that „this is an interesting, well-written story that provides additional insight into co-translational folding“, but raised two points that should be further investigated.

1. In absence of RAC, wt Ssb interacts with nascent chains whereas the mutant does not. This suggests that Ssb can also cooperate with other J-domain proteins. This aspect should be investigated further.

Authors: We are thankful that the reviewer brought up this important point, which we discuss now in more detail in the manuscript. The observation that wt Ssb is still able to interact with nascent chains in the absence of RAC might be due to two reasons: First, Ssb has an unusually high intrinsic ATPase activity in comparison to other Hsp70s (Lopez-Buesa *et al.*, 1998), meaning that it might be able to stably interact with substrates also in the absence of its ATPase stimulating co-factor RAC. Furthermore, substrate binding *per se* already enhances ATP-hydrolysis by Hsp70s (see e.g. Mayer & Kityk, 2015), which might explain residual functionality of wt Ssb in absence of RAC. A second explanation could be that Ssb is also functional with other Hsp40 co-chaperones. Although earlier experiments did not detect *in vitro* ATPase stimulation of Ssb by other yeast Hsp40s like Ydj1 or Sis1 (Lopez-Buesa *et al.*, 1998), it could be shown that overexpression of the ribosome-associated Hsp40 Jjj1 partially substitutes for Zuo1 in a RAC Δ strain (Meyer *et al.*, 2007; Albanese *et al.*, 2010). This suggests a possible cooperation of the Hsp70 Ssb with other ribosome-attached or cytosolic Hsp40 co-chaperones.

We now include these aspects in the discussion of the manuscript: "The residual functionality of Ssb observed in the absence of RAC might be due to two reasons which are not mutually exclusive: First, Ssb possesses an unusually high intrinsic ATPase activity in comparison to other Hsp70s (Lopez-Buesa *et al.*, 1998), which might allow efficient nascent chain binding also without the additional stimulation by RAC. A second explanation could be that Ssb is able to cooperate with other Hsp40 co-chaperones. Although earlier experiments did not detect *in vitro* ATPase stimulation of Ssb by other yeast Hsp40s like Ydj1 or Sis1 (Lopez-Buesa *et al.*, 1998), it could be shown that overexpression of the ribosome-associated Hsp40 Jjj1 partially substitutes for Zuo1 in a RAC Δ strain (Meyer *et al.*, 2007; Albanese *et al.*, 2010). This suggests a possible cooperation of the Hsp70 Ssb with other ribosome-attached or cytosolic Hsp40 co-chaperones in the absence of RAC."

2. A limitation of the manuscript is that the binding site of Ssb on the ribosome was not addressed.

Authors: We fully agree with the referee that mapping the ribosome-binding site(s) of Ssb is of highest interest and we started already to work on this important question prior to the referees' report. Unfortunately, up to now the data are not solid enough for publication. We need much more time to carefully control our results and thus the data cannot be incorporated into the revision of this manuscript.

However, to further support the importance of the two identified regions in Ssb as direct contact sites to ribosomes, we performed additional experiments, which we would like to add to the manuscript. We created three Ssa1 mutants that are all

FLAG-tagged: Wt-Ssa1, Ssa1 Δ C (corresponding to the Ssb1 Δ 601-13 mutant) and an Ssa1 version that contains the Ssb ribosome-binding elements (the Ssb KRR428 motif and 13 C-terminal Ssb residues). We can show the Ssa1 version that carries both ribosome-binding elements of Ssb exhibits enhanced ribosome binding, although it does not complement growth of an *ssb1,2* Δ strain.

We incorporated this new data as Supplementary Fig. 5 in the result section (please note: former Supplementary Figure 5 is now 6) and included the following paragraph: "To provide further evidence that the KRR-motif and the C-terminus of Ssb are directly involved in targeting to the ribosome, we created an N-terminal FLAG-tag version of Ssa1 (Ssa1_RB) carrying the two ribosome binding (RB) sites of Ssb1: residues 601-13 and the KRR-motif (Supplementary Fig. 5a). As controls we designed also a FLAG-tagged variant of wt Ssa1 or Ssa1 Δ C (corresponding to Ssb1 Δ 601-13). Expression of these Ssa1 versions in *ssb1,2* Δ cells did not complement growth under the conditions tested (Supplementary Fig. 5b). However, introduction of the two Ssb ribosome-binding regions enhanced binding of Ssa1_RB to ribosomes compared to wt Ssa1 and even more pronounced to Ssa1 Δ C (Supplementary Fig. 5c). Although also a small proportion of wt Ssa1 could be detected in the pellet, Ssa1_RB showed an enhanced migration with ribosomes in the sedimentation assay, which was reduced again under higher salt conditions (Supplementary Fig. 5d; note that the Ssb-specific antibody detects the Ssb1 residues in Ssa1_RB). We conclude that the two Ssb ribosome-binding regions promote targeting of other Hsp70s to the translation machinery suggesting that these Ssb regions directly interact with ribosomes."

Reviewers' Comments:

Reviewer #1 (Remarks to the Author):

The authors adequately addressed my concerns and I support publication of the current form of the manuscript.